# Structural Protein Analysis of Driver Gene Mutations in Conjunctival Melanoma

**DOI:** 10.3390/genes12101625

**Published:** 2021-10-15

**Authors:** Mak B. Djulbegovic, Vladimir N. Uversky, J. William Harbour, Anat Galor, Carol L. Karp

**Affiliations:** 1Bascom Palmer Eye Institute, University of Miami Miller School of Medicine, Miami, FL 33136, USA; mbd83@med.miami.edu (M.B.D.); harbour@med.miami.edu (J.W.H.); agalor@med.miami.edu (A.G.); 2Department of Molecular Medicine and USF Health Byrd Alzheimer’s Research Institute, Morsani College of Medicine, University of South Florida, Bruce B. Downs Blvd., MDC07, Tampa, FL 33612, USA; vuversky@usf.edu; 3Center for Molecular Mechanisms of Aging and Age-Related Diseases, Moscow Institute of Physics and Technology, Institutskiy Pereulok, 9, Dolgoprudny, 141700 Moscow, Russia; 4Ophthalmology, Miami Veterans Affairs Medical Center, Miami, FL 33136, USA; 5Research Services, Miami Veterans Affairs Medical Center, Miami, FL 33136, USA

**Keywords:** conjunctival melanoma, genetics, structural analysis, intrinsically disordered proteins, targeted therapies, mitogen-activated protein kinase (MAPK), Phosphatidylinositol-3-Kinase and Protein Kinase B (PI3K-akt), proto-oncogene B-Raf (BRAF), neuroblastoma v-ras oncogene homolog (NRAS), receptor tyrosine kinase (c-KIT), neurofibromatosis type 1 (NF1), phosphate and tensin homolog (PTEN)

## Abstract

In recent years, there has been tremendous enthusiasm with respect to detailing the genetic basis of many neoplasms, including conjunctival melanoma (CM). We aim to analyze five proteins associated with CM, namely BRAF, NRAS, c-KIT, NF1, and PTEN. We evaluated each protein for its intrinsically disordered protein regions (IDPRs) and its protein-protein interactions (PPI) with the Predictor of Natural Disordered Protein Regions (PONDR^®^) and the Search Tool for the Retrieval of Interacting Genes (STRING^®^). Our PONDR^®^ analysis found high levels of IDPRs in all five proteins with mutations linked to CM. The highest levels of IDPRs were in BRAF (45.95%), followed by PTEN (31.76%), NF1 (22.19%), c-KIT (21.82%), and NRAS (14.81%). Our STRING analysis found that each of these five proteins had more predicted interactions then expected (*p*-value < 1.0 × 10^−16^). Our analysis demonstrates that the mutations linked to CM likely affected IDPRs and possibly altered their highly complex PPIs. Quantifying IDPRs in BRAF, NRAS, c-KIT, NF1, and PTEN and understanding these protein regions are important processes as IDPRs can be possible drug targets for novel targeted therapies for treating CM.

## 1. Introduction

Conjunctival melanoma (CM) is an aggressive neoplasm of the ocular surface. While it is rare, the incidence of this neoplasm is on the rise [1,2]. The annual incidence of CM has been estimated at 0.2 to 0.8 cases per 1,000,000 people, and patient populations particularly susceptible to ultraviolet light damage such as non-Hispanic white men are at the highest risk of developing this malignancy [1,3,4,5]. CM may arise from primary acquired melanosis (PAM), nevus, or de novo [2,4,5]. While the origins of primary lesions and presenting symptoms vary, patients usually seek care because they notice a pigmented spot on their ocular surface. 

Wide excision with a “no touch technique” in combination with cryotherapy is the modern approach for managing CM [1,6,7,8,9,10,11]. Limitations of this technique include the concern for extensive insult to the ocular surface with excisional biopsy and the potential for residual neoplastic cells. To help reduce the rates of recurrence, most surgeons will add adjuvant brachytherapy and/or topical chemotherapy (mitomycin C, topical interferon alpha 2b) to the primary excision with cryotherapy [1,6,7,8,9,10,11,12,13,14,15,16,17]. Despite the use of adjuvant therapies, the cumulative rates of local recurrence at 10 years are 36.9% (CI: 26.5–49.9%) [6]. The origin of the CM lesion impacts the probability of metastatic spread and mortality. De novo mutations have the highest chance of metastasizing with 35% at 5 years and 49% at 10 years. These cases are associated with lower survivability as compared to the lesions that arise from PAM or nevus [18]. The burden of CM is evident as it has high rates of recurrence and mortality. 

In recent years, there has been a rise in precision medicine, and interest in the genetic basis of CM has blossomed. These molecular insights are crucial as they provide a potential understanding of the disease process, paving the road for novel medical therapeutics and targeted therapies. For example, patients with metastatic melanoma positive for the BRAF mutation can be treated with combined proto-oncogene B-Raf (BRAF) and mitogen-activated protein kinase kinase (MEK) inhibitors, and now they can also be treated with checkpoint inhibition through programmed cell-death 1 (PD-1) and cytotoxic T-lymphocyte antigen 4 (CTLA-4) receptors for patients that show systemic metastasis [5,19,20]. Large tumors harboring BRAF V600E mutations may benefit from neoadjuvant therapy with combined BRAF and MEK inhibitors followed by local excision as an alternative to exenteration [21]. KIT inhibitors and others may also be effective in the treatment of metastatic CM [2,22]. While there are targeted therapies that are indicated for the treatment of aggressive CM, recurrence and mortality rates remain high [6]. New therapeutics must be considered to reduce the burden of CM.

The development of novel molecular-based medical therapies provide a potential new avenue to treat CM. Novel therapies may possibly limit treatment related ocular surface damage and reduce recurrence, metastasis, and mortality rates. In a 2020 paper, Gkiala and Palioura highlighted the key genetic changes implicated in CM, where specific mutations were localized to the mitogen-activated protein kinase (MAPK) and phosphatidylinositol-3-kinase protein kinase B (PI3K-akt) pathways [23]. However, a gap remains regarding the impact the specific mutations identified have on five proteins involved in the MAPK and PI3k pathways. Specifically, it is unclear how these mutations impact the functional proteomics of these proteins and the interaction networks between the proteins. 

In the interest of advancing the current management of CM, it is important to detail the unknown components of this neoplasm’s proteome. While there are many components of the proteome to consider, an area in the field that is becoming increasingly linked to neoplastic development is intrinsically disordered proteins (IDPs) and hybrid proteins containing ordered domains and IDP regions (IDPRs). IDPs and IDPRs are proteins that contain global or local levels of non-3D structural components and provide a protein with a highly flexible and functional conformational ensemble [24,25,26]. Genetic mutations promote changes in IDPs and IDPRs and have been linked to the development of cancer [27]. Thus, IDPs and IDPRs are attractive candidates for novel therapeutics targeted at CM. IDPs, such as nuclear protein 1 (NUPR1), have been targeted to successfully treat melanoma cell lines [27]. Other IDPs and IDPRs may be suitable targets to target melanoma cell lines or other forms of melanoma. 

The impact that specific mutations in conjunctival melanoma have on IDPRs in proteins found in the MAPK and PI3K-akt pathways (BRAF, NRAS, c-KIT, NF1, and PTEN) has not been previously characterized. By utilizing a computational disorder-based structural analysis, we aim to bridge this gap. If intrinsically disordered proteins (IDPs) are present in proteins involved in the pathogenesis of CM, then they potentially provide a framework by which CM develops and a possible target for future drug development as IDPs have been successfully targeted to treat cancer [27]. Our analysis identified and intrinsically characterized disordered protein regions (IDPRs) and protein-protein interaction (PPI) networks of BRAF, NRAS, c-KIT, NF1, and PTEN. As such, this evaluation provides a framework for pathogenic molecular features of CM that can potentially be targeted for future therapeutic effects.

## 2. Materials and Methods

An initial literature search with key terms “conjunctival melanoma” and “genetics” was performed to understand the current genetics applications for the treatment of CM. Articles used in the literature search were collected from the National Library of Medicine MEDLINE databases. All forms of published scientific articles were considered, including original research, meta-analyses, and systematic reviews. All articles were limited to publications in the English language. A disorder-based structural analysis that utilized publicly available databases was conducted following the literature search (Figure 1). 

### 2.1. Pathways of Interest

The Kyoto Encyclopedia of Genes and Genomes (KEGG) is a database that houses high-throughput experimental biological data (available at: https://www.genome.jp/kegg/ (accessed on 15 May 2021)) [28,29,30]. One tool available on KEGG features interactive pathways, allowing for visualization and deeper understanding of the pathways of interest. The search terms used in KEGG were “mitogen active protein kinase” (MAPK) and “phosphatidylinositol-3-kinase protein kinase B” (PI3K-Akt). The purpose of viewing these pathways was to gain high-level understanding of their architecture. 

### 2.2. Protein Sequences 

The Universal Protein Resource (UniProt; available at: https://uniprot.org (accessed on 15 May 2021)) is a comprehensive database for protein data [31]. The search terms used were BRAF, NRAS, c-KIT, NF1, and PTEN, only the human variant of each gene product was selected, and the amino acid sequence in texted-based format (FASTA) for each human gene was obtained. If more than one human sequence is listed in UniProt for an entry, then the canonical sequence was selected. 

### 2.3. Structural Assessment

Structural propensity of each protein was analyzed. The structural propensity of each protein was analyzed. X-ray structures with the highest resolution (lowest Å) available on UniProt were evaluated for NF1 (https://www.uniprot.org/ (accessed on 15 May 2021)). The four proteins (BRAF, NRAS, c-KIT, and PTEN) were evaluated by AlphaFold2 [32], which is currently the most accurate computational method for predicting three-dimensional (3D) protein structures from the protein sequence.

### 2.4. Quantitative Disorder-Based Predictions

The five FASTA sequences used in this computational analysis (BRAF, NRAS, c-KIT, NF1, and PTEN) were run through the Predictor of Natural Disordered Protein Regions (PONDR^®^; available at: http://original.disprot.org/metapredictor.php (accessed on 10 June 2021)) and IUPred2A platform (https://iupred2a.elte.hu/ (accessed on 10 June 2021)). Both platforms are publicly available and represent tools that input a protein’s amino acid sequence and output quantitative, disorder-based data. In this study, we used four per-residue PONDR^®^ predictors including PONDR^®^ VLXT [33], PONDR^®^ VL3 [33], PONDR^®^ VSL2 [34], and PONDR^®^ FIT [35]. Two forms of IUPred2A [36] were used for the prediction of short and long disordered regions. A mean disorder profile (MDP) was also generated to assess average disorder prediction over all predictors used in this study.

### 2.5. Protein-Protein Interaction Network

The Search Tool for the Retrieval of Interacting Genes (STRING; available at: https://string-db.org/ (accessed on 10 June 2021)) [37] was used to generate detailed understanding of the functional interactions of the five identified gene products. All five FASTA sequences were input into the server, utilizing the same setting that included the highest confidence (0.900) and the maximum number of interactions possible (500).

## 3. Results

### 3.1. Pathways with Proteins of Interest

The MAPK signaling pathway (Kegg Entry ID: hsa04010; Figure 2) and the PI3K-Akt signaling pathway (Kegg Entry ID: hsa04151; Figure 3) show many different protein-protein interactions that promote cellular proliferation. The downstream effects of these pathways are made possible through protein-protein interactions (PPI) and any deviations in these interactions from normal can potentiate neoplastic change and promote tumor development. 

### 3.2. Protein Sequences

The protein sequences BRAF, NRAS, c-KIT, NF1, and PTEN (Appendix A) were used as primary inputs for PONDR^®^ and STRING analysis. Each unique protein sequence has mutations associated with CM (Table 1).

### 3.3. Structural Assessment

Specific mutations in BRAF, NRAS, c-KIT, NF1, and PTEN associated with CM (Table 1) are possible sources of aberrant PPI, which likely occur because of changes to structural properties. There are parts of all five proteins that show typical globular structure (Figure 4) possessing α-helices and β-pleated sheets. In addition to these highly structured domains, there is evidence of disordered segments that show a lack of inherent structure. These spaghetti-like entities are clearly observed in the AlphaFold2-generated structures of BRAF, NRAS, c-KIT, and PTEN (Figure 4). These regions of missing electron density correspond to IDPRs, which are segments that show high conformational flexibility and low propensity for structure and yet retain high levels of functionality. 

### 3.4. Quantitative Disorder Based Predictions

Our disorder-based computational analysis was aimed at quantifying the intrinsically disordered protein region (IDPR) identified in the structural assessment of each gene product. To begin, the IUPred2 (Short (S)) and IUPred2 (Long (L)), PONDR^®^ VLXT, PONDR^®^ VL3, PONDR^®^ VLS2, and PONDR^®^ FIT analysis (Figure 5) allowed for the visualization and confirmation of IDPRs within BRAF, NRAS, c-KIT, NF1, and PTEN. Furthermore, we quantified these IDPRs for each protein (Table 2). Based on the PONDR^®^ VSL2 outputs, BRAF had the highest levels of structural disorder BRAF (45.95%), followed by PTEN (31.76%), NF1 (22.19%), c-KIT (21.82%), and NRAS (14.81%). Therefore, using the established criteria for the classification of proteins based on their PPIDR scores where proteins are considered as highly ordered (PPDR < 10%), moderately disordered (10% ≤ PPDR < 30%), and highly disordered (PPDR ≥ 30%) [39], it was determined that NRAS, c-KIT, and NF1 belong to the category of moderately disordered proteins, whereas PTEN and BRAF are highly disordered.

### 3.5. Protein-Protein Interaction Network 

The next step in our investigation was focused on determining the significance of these structural findings. The presence IDPRs promotes conformational flexibility in each protein; therefore, these proteins likely do act as promiscuous binders and might have interaction partners outside of their well-defined roles in the MAPK and PI3K pathways. Our STRING analysis (Figure 6) demonstrated that each of the proteins that we analyzed has the capability of binding with many different partners. In fact, the number of interactors in the protein-protein interaction (PPI) network of BRAF, NRAS, c-KIT, NF1, and PTEN ranges from 60 to 327 (Table 3). The predicted number of interactions in the PPI network is highest for NRAS (7795), followed by PTEN (2297), BRAF (2213), NF1 (1790), and c-KIT (495). All of these values are significant (*p*-value < 10–16) as they vary greatly from their expected number of interactions. 

We also used STRING to investigate the intra-set interactions of human BRAF, NRAS, c-KIT, NF1, and PTEN. The analysis revealed that these proteins are heavily linked by binary protein-protein interactions. The intra-set networks of all five proteins were connected by 14 protein-protein interactions. Since the expected number of interactions among proteins in a similar size set of proteins randomly selected from human proteome is equal to two, this internal PPI network of human CM-related proteins has significantly more interactions than expected, being characterized by a PPI enrichment *p*-value of 1.4 × 10^−9^.

## 4. Discussion

BRAF, NRAS, c-KIT, NF1, and PTEN have specific mutations (Table 1) that are associated with CM, and each mutation likely alters the functionality of these proteins. Our structural analysis confirmed the presence of intrinsically disordered protein regions (IDPRs) in all these key proteins (BRAF (45.95%) > PTEN (31.76%) > NF1 (22.19%) > c-KIT (21.82%) > NRAS (14.81%)). Furthermore, STRING analysis demonstrated highly complex protein-protein interaction (PPI) networks as every network had many predicted interactions beyond their expected binding behavior (*p*-value < 1.0 × 10^−16^). These findings suggest that mutations associated with these proteins likely have an impact on protein interactions, are secondary to abhorrent regulation of IDPRs, and possibly promote oncogenesis and neoplastic development. This is consistent with findings that mutations can affect IDPRs and promote oncogenesis [40]. 

To our knowledge, this is the first study to analyze the mutations associated with the pathogenesis of CM, namely for their IDPRs and the PPI network of each of these proteins. In recent years, there has been a push to target intrinsically disordered protein regions involved in various cancers including adenocarcinoma, Ewing’s sarcoma, and lymphoma [27,41,42]. While targeting IDPRs is challenging as they are highly dynamic and have poorly defined structures, intrinsically disordered protein (IDP) drug discovery campaigns [43] are addressing these limitations. In pancreatic ductal adenocarcinoma, an IDP, nuclear protein 1 (NUPR1), has been successfully targeted in vivo to completely arrest development of the cancer in mice [44]. These advances in IDP-based drug discovery methods may provide a viable target for future campaigns targeting molecular features of CM, which include the high level of IDPRs in BRAF, NRAS, c-KIT, NF1, and PTEN. Hopefully, the extensive intrinsic disorder will improve the chances of success, potentially yielding novel therapeutics to treat CM. 

There are already several different drugs that are available for the extraocular melanomas, such as BRAF inhibitors and c-KIT inhibitors, and it is possible repurpose them to treat CM [45,46,47]. These therapies are helping the management of melanomas, but they do not always induce remission. Therefore, other targets, such as IDPRs in BRAF, NRAS, c-KIT, NF1, and PTEN, could be considered. There are likely compounds that are commercially available that may be able to target the highly dynamic IDPRs of any of our five proteins of interest. 

Our analysis found high levels of IDPR’s in BRAF, NRAS, c-KIT, NF1, and PTEN and demonstrates the presence of highly complex PPIs, which may be possible future therapeutic targets. The presence of IDPRs is important as the solution space for potential drug targets is likely significantly larger than previously thought. CM’s mutational profile will likely consist of one or more the of these mutations. If more than one mutation is present, combination-based medical therapy may be warranted in order to target the neoplasm at multiple points in its protein interaction network. 

As with all studies, this study had limitations. The databases and tools utilized in this analysis do not provide a comprehensive nor exhaustive representation of the proteomics involved in CM. As with all bioinformatics investigations, translations to basic science and clinical outcomes are needed. The tools used herein revealed the molecular features of five key proteins related to CM. 

To our knowledge, this is the first study to identify IDPRs in five important proteins that have been linked to the pathogenesis of CM. There is evidence to suggest that targeting IDPRs play a role in cancer treatments [27] and the finding in this study suggest a possible avenue for studying new treatment targets for CM. These findings are critical for advancing the management of this potentially lethal neoplasm as it may allow for the development of new therapeutics that target these structural properties. If such targeting comes to fruition, perhaps other tumors with these proteins and IDPRs could possibly be therapeutically targeted. Further research will be needed in order to translate these genomic and proteomic based discoveries into tangible improvements of patient outcomes.

## 5. Conclusions

Neoplastic lesions are increasingly treated at a molecular level, and one method to address specific mutational proteomic variants is through targeted therapy. Potentially, drug therapies targeted at intrinsically disordered protein regions of BRAF, NRAS, c-KIT, NF1, and PTEN offer a possible avenue to new treatments that can aid in the management of this neoplasm. Ultimately, translating these findings to the clinical management of these patients will require an integrative omics approach. The hope is that small molecule inhibitors and targeted therapies aimed at IDPRs may assist in the management of conjunctival melanoma.

## Figures and Tables

**Figure 1 genes-12-01625-f001:**
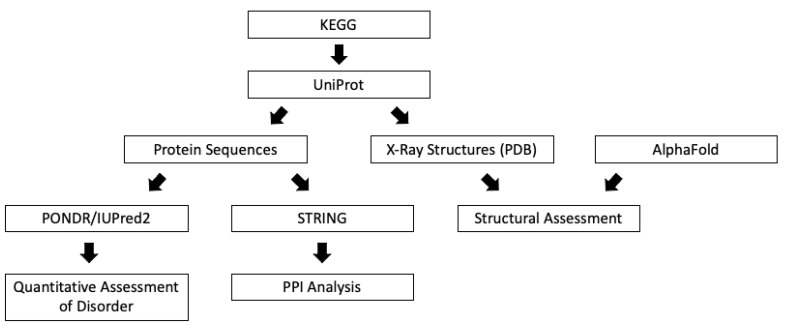
A flowchart of structural analysis of driver genes mutations in conjunctival melanoma. KEGG; Kyoto Encyclopedia of Genes and Genomes. UniProt; Universal Protein resource. PONDR, Predictor of Natural Disordered Regions. IUPred2; web server for the prediction of intrinsically unstructured regions of proteins. STRING; Search Tool for the Retrieval of Interacting Genes. PPI; Protein-Protein Interaction. PDB; Protein Data Bank. AlphaFold; Artificial Intelligence Program for Protein Structure Prediction.

**Figure 2 genes-12-01625-f002:**
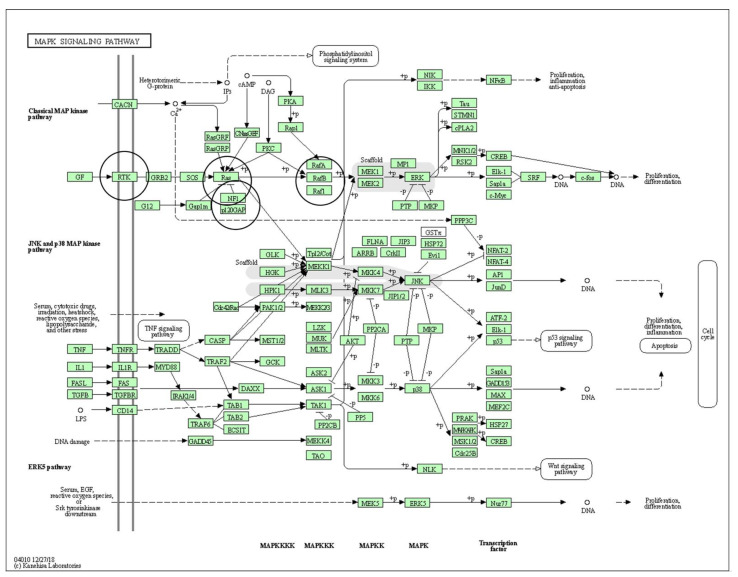
Refs. [29,30,38]. KEGG Pathway–Mitogen-activated protein kinase (MAPK; KEGG entry ID: hsa04010) pathways. The classical MAPK pathway is involved in conjunctival melanoma (CM). The black circles identify the proteins with known mutations in CM, c-kit (map label: RTK), NRAS (map label: NRAS), NF1 (map label: NF1), and BRAF (map label: RafB). More information about any protein seen in the pathway (i.e. abbreviation definition, gene function, protein sequence, etc.) can be accessed through the interactive Kyoto Encyclopedia of Genes and Genomes (KEGG) pathway (available at: https://www.genome.jp/pathway/hsa04010 (accessed on 15 May 2021)).

**Figure 3 genes-12-01625-f003:**
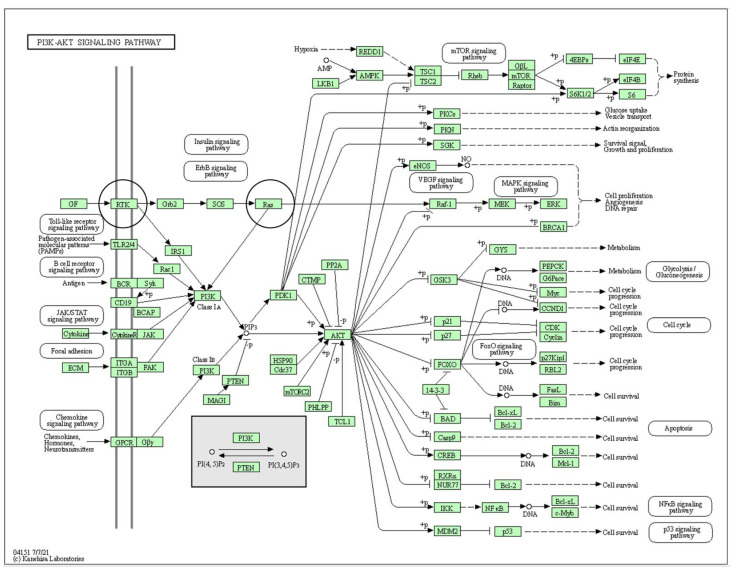
References [29,30,38]. KEGG Pathway—Phosphatidylinositol-3-kinase protein kinase B (PI3K-akt; KEGG entry ID: hsa04151). The classic PI3K-akt pathway is involved in conjunctival melanoma (CM). The black circles identify proteins with known mutations in CM, c-KIT (map id: RTK), and NRAS (map id: Ras). More information about any protein seen in the pathway (i.e. abbreviation definition, gene function, protein sequence, etc.) can be accessed through the interactive Kyoto Encyclopedia of Genes and Genomes (KEGG) pathway (available at: https://www.genome.jp/pathway/hsa04010 (accessed on 15 May 2021)).

**Figure 4 genes-12-01625-f004:**
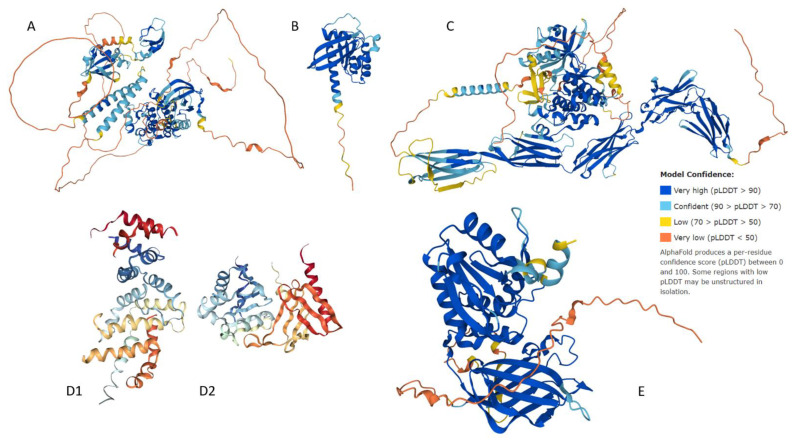
Universal Protein Resource (UniProt) structural output for human (**A**) BRAF, (**B**) NRAS, (**C**) c-KIT, (**D1**,**D2**) NF1, and (**E**) PTEN. Structures for BRAF, NRAS, c-KIT, and PTEN were generated by AlphaFold2, whereas structure of human NF1 is presented by X-ray crystal structures of its two domains from the protein data bank. Confidence of models generated by AlphaFold2 is color coded (see legend in the figure).

**Figure 5 genes-12-01625-f005:**
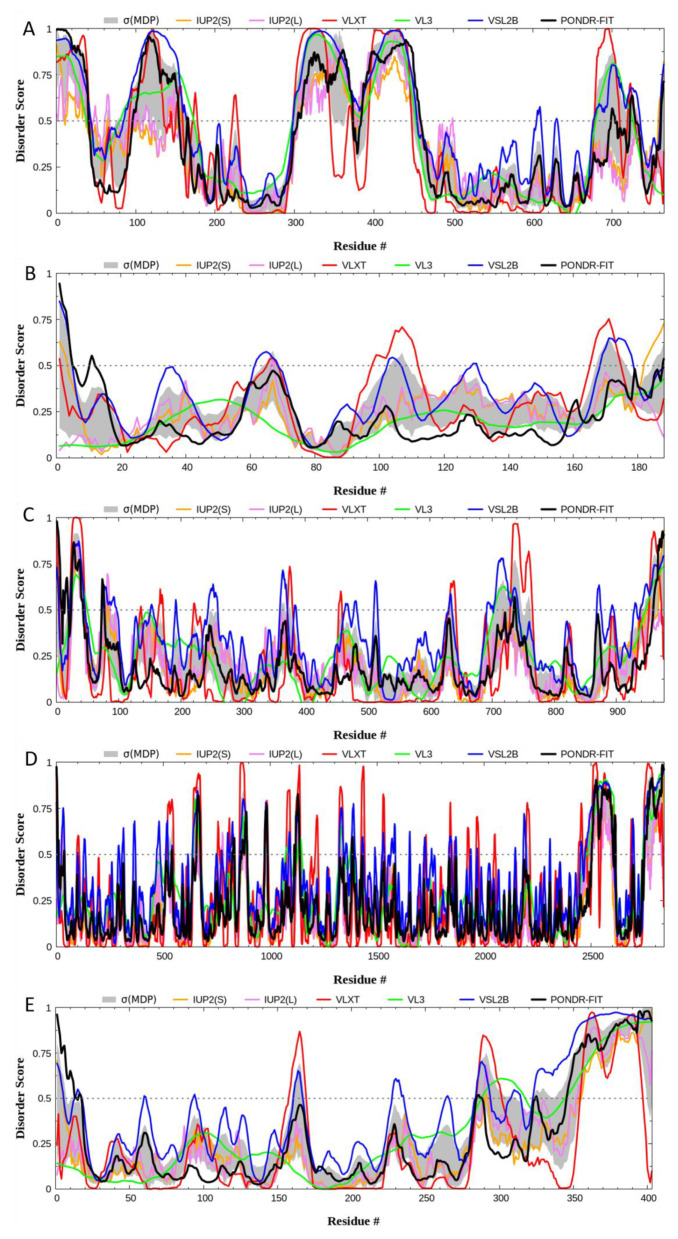
Per-residue intrinsic disorder profiles generated for human BRAF (**A**), NRAS (**B**), c-KIT (**C**), NF1 (**D**), and PTEN (**E**) by mean disorder profile (MDP) and by well-known disorder predictors, including IUPred2 (Short (S)) and IUPred2 (Long (L)), PONDR^®^ VLXT, PONDR^®^ VL3, PONDR^®^ VLS2, and PONDR^®^ FIT. The outputs of the evaluation of the per-residue disorder propensity by these tools are represented as real numbers between one (ideal prediction of disorder) and zero (ideal prediction of order). A threshold of 0.5 was used to identify disordered residues and regions in query proteins. All five proteins show extensive intrinsic disorder, and BRAF’s quantity of intrinsic disorder was the highest at 36.73% by average disorder predictions.

**Figure 6 genes-12-01625-f006:**
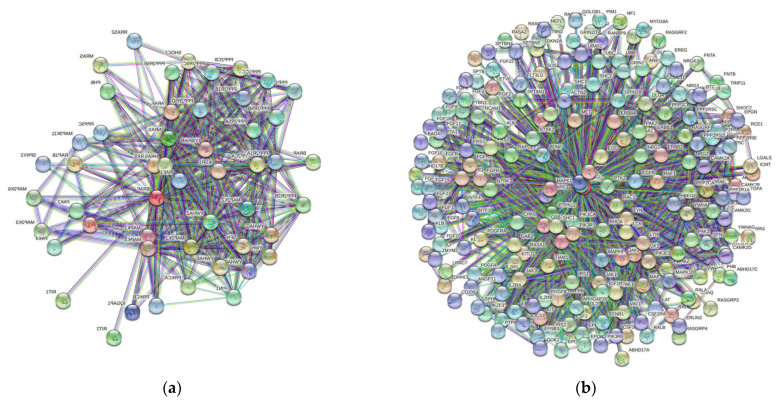
Search Tool for the Retrieval of Interacting Genes (STRING) output for (**a**) BRAF, (**b**) NRAS, (**c**) c-KIT, (**d**) NF1, and (**e**) PTEN. This analysis shows multiple proteins (circles) and their extensive interaction network (lines) for the five proteins involved in the pathogenesis of conjunctival melanoma. This demonstrates very complex protein binding ability beyond the MAPK and PI3-akt pathways. The ability of these proteins to interact in an extensive interaction network is possible through intrinsically disordered protein regions.

**Table 1 genes-12-01625-t001:** Genes with mutations known to be associated with the development of conjunctival melanoma.

Gene Name	Known Gene Mutations	UniProt ID	Structure Source
*BRAF*	G469A, D594G, V600E/K/R	P15056	AlphaFold2
*NRAS*	Q12C, Q13D, Q61R/H/L/K	P01111	AlphaFold2
*c-KIT*	*	P10721	AlphaFold2
*NF1*	More than 25 distinct mutations	P21359	1NF1and 3PEG (PDB IDs)
*PTEN*	Chromosome 10q deletion	P60484	AlphaFold2

* Exact mutation has not been noted in the literature. UniProt; Universal Protein resource. PDB; Protein Data Bank.

**Table 2 genes-12-01625-t002:** Predictor of Natural Disordered Protein Region (PONDR) analysis quantifying the percentage of predicated intrinsic disorder of human proteins associated with conjunctival melanoma.

	Gene Name				
Predictor	*BRAF*	*NRAS*	*c-KIT*	*NF1*	*PTEN*
PONDR^®^ VLXT	33.55%	14.29%	12.30%	19.90%	17.62%
PONDR^®^ VL3	43.08%	0.00%	9.53%	13.46%	22.08%
PONDR^®^ VSL2	45.95%	14.81%	21.82%	22.19%	31.76%
Average	40.86%	9.70%	14.55%	18.52%	23.82%

**Table 3 genes-12-01625-t003:** Search Tool for the Retrieval of Interacting Genes (STRING) analysis with mutations known to be associated with the development of conjunctival melanoma.

Gene Name	Proteins in Network	Expected Number of Interactions	Predicted Number of Interactions	*p*-Value
*BRAF*	109	253	2213	<10^−16^
*NRAS*	327	1619	7795	<10^−16^
*c-KIT*	60	177	495	<10^−16^
*NF1*	90	388	1790	<10^−16^
*PTEN*	157	622	2297	<10^−16^

*p*-value; PPI enrichment *p*-value determined by difference between expected and predicted number of interactions.

## Data Availability

The data presented in this study are available on publicly accessible repositories. The data presented in this study are openly available in the Kyoto Encyclopedia of Genes and Genomes at https://www.genome.jp/kegg/ (accessed on 15 May 2021), the Protein Data Bank at https://www.rcsb.org/ (accessed on 15 May 2021), and the Universal Protein Resource at https://uniprot.org (accessed on 15 May 2021). Protein pathways, structures, and sequences were sourced from these databases and used in subsequent analysis.

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
