# Peer review of "Structural Protein Analysis of Driver Gene Mutations in Conjunctival Melanoma"

_genes, 2021, doi:10.3390/genes12101625_

Round 1

Reviewer 1 Report

In this work, Djulbegovic et al evaluated 5 proteins that are associated with conjunctival melanoma using several (online) tools. They report on the intrinsically disordered protein regions (IDPR) and its protein-protein interactions (PPI). While the content is absolutely novel, and clearly written and presented, the rationale for this study is somewhat difficult to understand. Yes the reported genes are relevant for conjunctival melanoma, but the findings are based on non-conjunctival  material (as far as I understand), and only a theoretical claim is presented that future therapies could be aimed at these proteins. I would suggest to either rewrite the paper to a more general introduction into the relevance of IDPRs and PPI's for ocular oncologists (perhaps using conj mel as an example), or to provide the (currently missing) link between reported proteins and upcomming / suggested therapy.

Reviewer 2 Report

The article by Djulbegovic et al. reports clearly, precisely and with absolute clinical application information about conjunctival melanoma. I appreciated the translational research and I suggest improving the clinical application. Figures are adapted and just some points needed to be improved.

  • Figure 1, correct caption
  • Material and method: provide a flowchart of your database research.
  • Figures 1 and 2 are well-presented
  • Add targeted therapies: read and cite

- Sun, J., Carr, M. J., & Khushalani, N. I. (2020). Principles of Targeted Therapy for Melanoma. The Surgical clinics of North America100(1), 175–188. https://doi.org/10.1016/j.suc.2019.09.013

Rossi, E., Maiorano, B. A., Pagliara, M. M., Sammarco, M. G., Dosa, T., Martini, M., Rindi, G., Bria, E., Blasi, M. A., Tortora, G., & Schinzari, G. (2019). Dabrafenib and Trametinib in BRAF Mutant Metastatic Conjunctival Melanoma. Frontiers in oncology9, 232. https://doi.org/10.3389/fonc.2019.00232

Demirci, H., Demirci, F. Y., Ciftci, S., Elner, V. M., Wu, Y. M., Ning, Y., Chinnaiyan, A., & Robinson, D. R. (2019). Integrative Exome and Transcriptome Analysis of Conjunctival Melanoma and Its Potential Application for Personalized Therapy. JAMA ophthalmology137(12), 1444–1448. https://doi.org/10.1001/jamaophthalmol.2019.4237

  • Line 148-150 report very fascinating items. Did you find the same IDPRs related to other tumours (or from the ectodermal structures)?
